# Medical Vision-Language Pretraining through Contrastive Learning of Positive and Negative Mention

## Abstract

In recent years, contrastive learning techniques have achieved significant success and have been widely applied in both general and medical domains. In the general domain, image captions typically describe only objects present in the image. However, in the medical field, radiology reports contain both sentences confirming the presence of diseases or abnormalities (positive mentions) and sentences explicitly ruling them out (negative mentions). Current vision-language pretraining models in the medical domain often overlook this critical distinction in both model evaluation (e.g., zero-shot classification) and training processes. In this paper, we suggest adding a zero-shot classification evaluation method. Unlike previous approaches that only assess the semantic similarity between medical images and positive mentions of different disease categories, this method evaluates the model's ability to distinguish between medical images and both positive and negative mentions of given disease category. Furthermore, to better capture the complex semantic relationships between medical images and the corresponding radiology reports, we introduce a visual entailment based contrastive learning method, explicitly modeling the entailment, contradiction, and neutral relationships between medical images and report sentences. Experimental results demonstrate that integrating this new evaluation method provides a more comprehensive evaluation of vision-language pretraining models in the medical domain. Additionally, our model achieves state-of-the-art performance across various downstream tasks, highlighting the effectiveness of our approach.

## 1 Introduction

Attributed to the rapid development of deep learning, an increasing number of medical tasks can be accomplished by deep learning models, such as classification Liskowski & Krawiec (2016); Fu et al. (2016), segmentation Yang et al. (2018); Zhang et al. (2018), and report generation Zeng et al. (2020); Wu et al. (2023b). However, achieving acceptable performance often requires task-specific annotated data, which depends on laborious and expensive labeling by clinical experts, posing a challenge and being time-consuming. To reduce reliance on annotated data, researchers have introduced self-supervised training methods, with contrastive learning Chen et al. (2020); Radford et al. (2021) being the most typical. When training with contrastive learning methods, the model first undergoes self-supervised training on a large amount of unannotated data, and then fine-tunes with a small amount of annotated data on downstream tasks, achieving the same satisfactory level of performance. The application of contrastive learning in medical tasks has effectively alleviated the model's dependence on expensive annotated data.

However, there is a critical distinction between general and medical domains image-text pairs, as illustrated in Figure 1. In general domain, image-text pairs almost only contain positive mentions. The content described in the text usually appears in the images Radford et al. (2021). But in medical domain, except positive mentions, image-report pairs also contain a large amount of negative mentions, where sentences in report explicitly ruling out the category not appearing in image. Current vision-language pretraining models in the medical domain often overlook this critical distinction in both model evaluation and training processes. This leads to limitations in evaluating model performance and can result in defects in downstream tasks.

Based on this issue we find, we suggest adding an additional evaluation method, which we call Positive-Negative Contrastive (PNC) evaluation method. As a contrast, we refer to the previous evaluation mode as the Positive-Only Similarity (POS) evaluation method. In the POS evaluation method, metrics reflect the model's ability to distinguish only the semantic similarity between medical images and positive mentions of different disease categories. However, in PNC evaluation method, metrics reflect the performance of models in distinguishing semantic similarity between medical images and both positive and negative mentions of given disease category. Through experiments we find, most models experience varying degrees of decline in their metrics in PNC evaluation method, indicating that these models have varying degrees of defects in learning the distribution of image and text features.

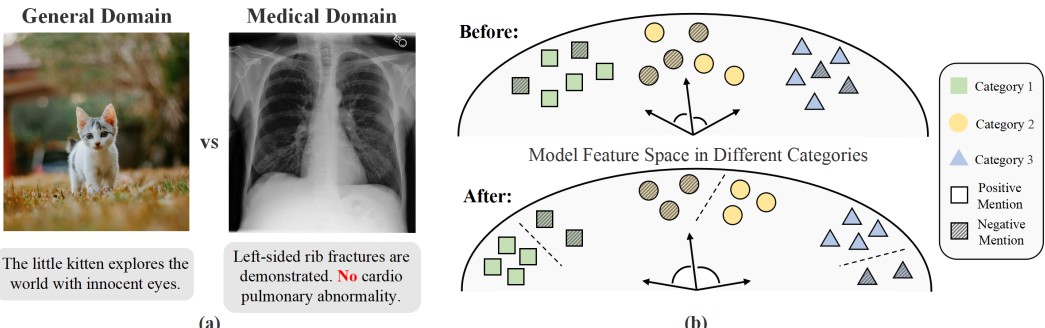

Figure 1: In general domain, image-text pairs almost only contain positive mentions. But in medical domain, image-report pairs usually contain both positive mentions and negative mentions. In our method, model can not only better distinguish semantic similarity between medical images and positive mentions of different disease categories ("inter-class similarity") but also better distinguish semantic similarity between medical images and both positive and negative mentions of given disease category ("intra-class similarity"). **(a)** Distinction between general and medical domains image-text pairs. **(b)** The change in the distribution of the model feature space in our method.

To address this issue, we introduce an innovative training method. By drawing on the ideas of the Visual Entailment task Xie et al. (2019),we consider both the positive and negative mentions during the contrastive learning process, introducing the **V**isual **E**ntailment based **C**ontrastive **L**earning (VECL) method, which explicitly modeling the entailment, contradiction, and neutral relationships between medical images and report sentences. Additionally, to adapt model training, we modified the classic contrastive learning loss function InfoNCE Oord et al. (2018), expanding it from two dimensions to three. To construct visual entailment relationships, we need additional supervisory signals. Fortunately, large language models (LLMs) have shown strong text understanding and inductive reasoning capabilities OpenAI (2023), allowing us to use LLM to extract disease category labels of report sentences. Then based on a specific generation rule, we use these labels to consturct visual entailment relationships among all samples within a batch and construct the similarity matrix label. Code and models are available at [1].

In summary, our contributions can be summarized as follows:

- **Suggestion of the Positive-Negative Contrastive (PNC) Evaluation Method.** Experimental results demonstrate that integrating PNC evaluation method provides a more comprehensive evaluation of vision-language pretraining models in the medical domain.
- **Introduction of the Visual Entailment Based Contrastive Learning (VECL) Method.** We integrate the Visual Entailment task into contrastive learning process to model the entailment, contradiction, and neutral relationships between medical images and report sentences.
- **Achieving SOTA Performance Across Various Downstream Tasks.** We compared different baseline models on various downstream tasks, and the experimental results demonstrate that we surpassed all baseline models on all metrics for all tasks.

---

[1]https://anonymous.4open.science/r/ICLR2025-92A0/readme

## 2 METHOD

In this section, we first illustrate the Positive-Negative Contrastive (PNC) evaluation method, and then describe the framework of the Visual Entailment Based Contrastive Learning (VECL) method. The framework of VECL method is shown in Figure 4.

### 2.1 POSITIVE-NEGATIVE CONTRASTIVE EVALUATION METHOD

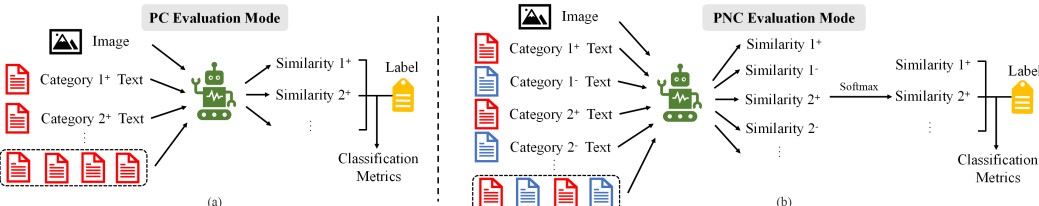

Figure 2: The difference between previous POS evaluation method and our PNC evaluation method. In PNC evaluation method, both positive and negative mentions of different disease categories are used as text inputs to compute similarities. Finally, after passing through the softmax, the similarities are sent to calculate the metrics with labels.

As shown in Figure 2, the difference between previous POS and our PNC evaluation method is in model inference phase. In POS evaluation method, only positive mentions of different disease categories are used as text inputs, designated as "There is {disease}", which is subsequently used to calculate similarities with images and directly with labels to compute classification metrics. But in PNC evaluation method, both positive and negative mentions of different disease categories are used as text inputs, designated as "There is {disease}" and "There is no {disease}". After calculating the similarity between all texts and images, the similarities of positive and negative mentions within the same category are normalized by softmax, and then the normalized scores of all categories' positive mentions are used to calculate classification metrics with labels.

In the PNC evaluation method, the model is evaluated based on the distance between positive images and positive text in the feature space, as well as the distance between negative sample images and negative sample text, given a disease category. As shown in Figure 3, the metric scores reach a satisfactory level only when the model can bring positive images closer to positive text and farther from negative text, while simultaneously bringing negative images closer to negative text and farther from positive text in the feature space.

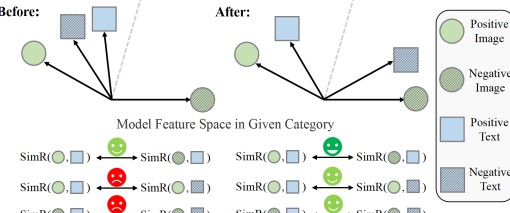

Figure 3: The change in the distribution of the model's feature space in our method. In our method, the model can better distinguish between positive and negative mentions.

### 2.2 AUTOMATIC LABEL EXTRACTION MODULE

In the automatic label extraction module, the report is first segmented into sentences by an LLM, and any sentence unrelated to radiological diagnosis is filtered out. The filtered sentences are then sent back into the LLM to extract the corresponding labels. Before label extraction, we selected 24 lung disease categories from common chest X-ray datasets, along with a 25th category representing other diseases or no abnormality, to form the label categories set $C$.

$$C = \{1^+, 1^-, 2^+, 2^-, \ldots, 24^+, 24^-, 25\} \tag{1}$$

With the label categories set, we can extract the labels of report sentences by LLM. Assuming the sentence $i_n$ in report $i$ is related to disease category $r$, the label output by the LLM is denoted as $C_i^n$. If the sentence $i_n$ in report $i$ is positive mention of disease category $r$, then $C_i^n = r^+$; If the sentence $i_n$ in report $i$ is negative mention of disease category $r$, then $C_i^n = r^-$; If the disease category is other diseases or no abnormality founding, then $C_i^n = 25$. We refer to $r^+$ and $r^-$ as each other's opposite label, and the opposite label of $25$ is itself. If a report sentence contains multiple diseases, a list of labels will be generated. In all subsequent processing, the list of labels containing multiple

diseases will be split into individual labels for handling. According to these rules, labels can be extracted from the report sentences in the entire training set.

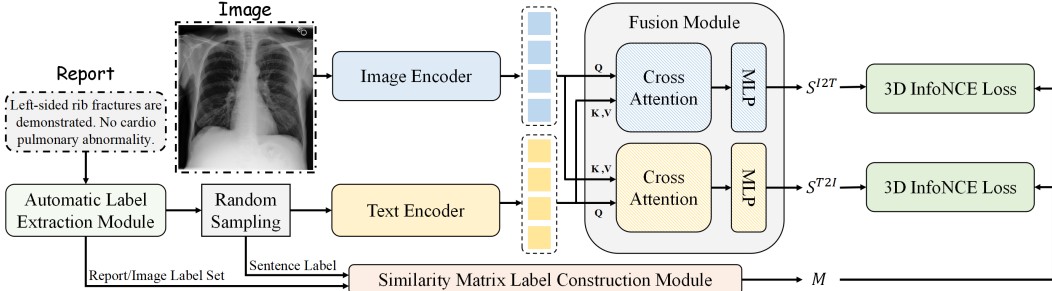

Figure 4: The framework of VECL method. The image enters the image encoder, while the text first goes through the automatic label extraction module to obtain the segmented report sentences and their corresponding sets of image labels. The segmented report sentences are then sampled and fed into the text encoder. After encoding, the image features and text features are processed through a fusion module to calculate similarities, resulting in $S^{I2T}$ and $S^{T2I}$. Meanwhile, the similarity matrix label construction module builds a similarity label matrix $M$ based on the image label set and the label of the sampled report sentence. Finally, $S^{I2T}$ and $S^{T2I}$ are used with M to calculate the loss using the 3D InfoNCE loss function.

## 2.3 ENCODER AND FUSION MODULE

For encoders, only one sentence from complete report is random sampled as the input of text encoder, while the complete image after data augmentation is the input of image encoder. Assume that $x_i$ represents the $i$-th image in training set and $y_j$ represents the $j$-th report in training set, $\Phi_I$, $\Phi_T$ and $\Phi_F$ represent the image encoder, the text encoder and the fusion module, respectively. So the intermediate results and the image-text similarity produced by the model's forward process are as follows:

$$x_i^I = \Phi_I(x_i), \ y_j^T = \Phi_T(y_j) \tag{2}$$

$x_i^I$ and $y_j^T$ represent the image features output by the image encoder and the text features output by the text encoder, respectively.

$$S_{ij}^{I2T} = \Phi_F(x_i^I, y_j^T) = MLP(CrossAtt(Q = x_i^I, K = y_j^T, V = y_j^T)) \tag{3}$$

$$S_{ji}^{T2I} = \Phi_F(y_j^T, x_i^I) = MLP(CrossAtt(Q = y_j^T, K = x_i^I, V = x_i^I)) \tag{4}$$

$S_{ij}^{I2T}$ and $S_{ji}^{T2I}$ represent the image-text similarity output by the fusion module, with each serving as the query in the cross attention process, respectively. Here $S_{ij}^{I2T}$ and $S_{ji}^{T2I}$ both are a one-dimensional vector, where $S_{ij}^{I2T}, S_{ji}^{T2I} \in \mathbb{R}^3$. For any image $x_i$ and any report $y_j$ within a batch, the image-text similarities between them ultimately form two similarity matrices $S^{I2T}$ and $S^{T2I}$, where $S^{I2T}, S^{T2I} \in \mathbb{R}^{N \times N \times 3}$. Here N is the batch size.

## 2.4 SIMILARITY MATRIX LABEL CONSTRUCTION MODULE

Similar to model's forward process, only one label of the sampled report sentence as text label and all labels of a complete report as the image label set are sent into similarity matrix label construction module. In this module, we integrate the visual entailment task into contrastive learning process to model the entailment, contradiction, and neutral relationships between medical images and report sentences. Specifically, we assess the relationship between the text label, opposite label of the text label, and the image label set. If the image label set includes the text label, we consider the relationship between the image and the report sentence to be entailment. If the image label set includes the opposite label of the text label, we consider the relationship between the image and the report sentence to be contradiction. If the image label set neither includes the text label nor its opposite label, we consider the relationship between the image and the report sentence to be neutral. For report sentences that contain multiple labels, the entire sentence is only considered to satisfy the entailment relationship if all the sub-labels satisfy the entailment relationship. If any sub-labels are in a contradiction relationship, the entire sentence is considered to be in a contradiction relationship.

We use three basis vectors:$[1,0,0]$, $[0,1,0]$, and $[0,0,1]$ to represent these relationships, respectively. Specifically, when the text label is 25, we consider the relationship between the report sentence and any image to be neutral.

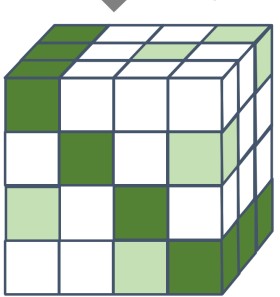

| | | $C_{i_1}$ | $C_{i_2}$ | $C_{i_{N-1}}$ | $C_{i_N}$ |
|---|---|---|---|---|---|
| $C_{j_1}^{k_1}$ | $\neg C_{j_1}^{k_1}$ | [1,0,0] | [0,1,0] | ⋯ [0,0,1] | [0,1,0] |
| $C_{j_2}^{k_2}$ | $\neg C_{j_2}^{k_2}$ | [0,1,0] | [1,0,0] | ⋯ [1,0,0] | [1,0,0] |
| ⋮ | | ⋮ | ⋮ | ⋮ | ⋮ |
| $C_{j_{N-1}}^{k_{N-1}}$ | $\neg C_{j_{N-1}}^{k_{N-1}}$ | [0,1,0] | [0,0,1] | ⋯ [1,0,0] | [0,0,1] |
| $C_{j_N}^{k_N}$ | $\neg C_{j_N}^{k_N}$ | [0,1,0] | [1,0,0] | ⋯ [0,1,0] | [1,0,0] |

Similarity Matrix Label $M$

1: **Input:**
2:   The batch data $B$, $len(B) = N$
3:   For any report (image) $i$, its labels set $C_i$ and row in the batch $R_i$
4:   For any sentence $j_k$ in report $j$, its label $C_j^k$, opposite label $\neg C_j^k$ and column in the batch $L_j$
5: **Output:**
6:   The similarity matrix label $M$, $shape(M) = [N, N, 3]$
7: **for** $i$ **in** $B$ **do**
8:     **if** $C_j^k$ **in** $C_i$ **then**
9:         $M(R_i, L_j) = [1, 0, 0]$
10:    **else if** $\neg C_j^k$ **in** $C_i$ **then**
11:        $M(R_i, L_j) = [0, 0, 1]$
12:    **else**
13:        $M(R_i, L_j) = [0, 1, 0]$
14:    **end if**
15: **end for**

Figure 5: The algorithm for constructing similarity matrix labels $M$. Based on the assessment of the relationship between the image label set and the label of the report sentence as well as its opposite label, the relationships are represented using $[1, 0, 0]$, $[0, 1, 0]$, and $[0, 0, 1]$ for entailment, contradiction, and neutral relationships, respectively.

Figure 6: The schematic diagram of constructing the similarity matrix labels. Within a batch, each image and report sentence is matched pairwise to form a relationships vector, and these vectors, which are number of batch size × batch size, are concatenated to construct the similarity matrix label $M$.

With the vectors representing relationships between image and report sentence, we can construct the similarity matrix label $M$, where $M \in \mathbb{R}^{N \times N \times 3}$. The detiled rules to assess the relationship between image and report sentence and construct the similarity matrix label $M$ are shown in Figure 5 and Figure 6. Thought these rules, we obtain the similarity matrix label $M$ that considers entailment, contradiction, and neutral relationships.

## 2.5 3D INFONCE LOSS

Let $d$ represent the position along the third dimension, where $d \in \{1, 2, 3\}$, $M(d)$, $S^{I2T}(d)$ and $S^{T2I}(d)$ represent the slices along the third dimension of $M$, $S^{I2T}$ and $S^{T2I}$ at positions $d$, respectively. For the given $d$, $M(d)$, $S^{I2T}(d)$ and $S^{T2I}(d)$ are each a two-dimensional matrix so can be incorporated in previous 2D InfoNCE loss $\mathcal{L}(d)$, which contains image-to-text alignment item $\mathcal{L}^{I2T}(d)$ and text-to-image alignment item $\mathcal{L}^{T2I}(d)$. And for both $\mathcal{L}^{I2T}(d)$ and $\mathcal{L}^{T2I}(d)$, each includes a cross-entropy loss calculated along the row-wise direction and the column-wise direction as follows:

$$\mathcal{L}_0^{I2T}(d) = -\frac{1}{N} \sum_{i=1}^{N} \sum_{j=1}^{N} \underset{dim=0}{\text{Norm}}(M_{ij}(d)) \cdot \log(\underset{dim=0}{\text{Softmax}}(e^{S_{ij}^{I2T}(d)})) \tag{5}$$

$$\mathcal{L}_1^{I2T}(d) = -\frac{1}{N} \sum_{j=1}^{N} \sum_{i=1}^{N} \underset{dim=1}{\text{Norm}}(M_{ij}(d)) \cdot \log(\underset{dim=1}{\text{Softmax}}(e^{S_{ij}^{I2T}(d)})) \tag{6}$$

Here, Norm() and Softmax() refers to the normalization function, $dim = 0$ and $dim = 1$ under them refer to normalization on row-wise and column-wise, respectively.

$$\underset{dim=0}{\text{Norm}}(M_{ij}(d)) = \begin{cases} \frac{M_{ij}(d)}{\sum_{i=1}^{N} M_{ij}(d)} & \text{if } \sum_{i=1}^{N} M_{ij}(d) \neq 0, \\ 0 & \text{if } \sum_{i=1}^{N} M_{ij}(d) = 0 \end{cases} \tag{7}$$

Similarly, we can compute $\mathcal{L}_0^{T2I}(d)$, $\mathcal{L}_1^{T2I}(d)$ and obtain $\mathcal{L}^{I2T}(d)$, $\mathcal{L}^{T2I}(d)$, $\mathcal{L}(d)$.

$$\mathcal{L}^{I2T}(d) = \mathcal{L}_0^{I2T}(d) + \mathcal{L}_1^{I2T}(d) \tag{8}$$

$$\mathcal{L}^{T2I}(d) = \mathcal{L}_0^{T2I}(d) + \mathcal{L}_1^{T2I}(d) \tag{9}$$

$$\mathcal{L}(d) = \mathcal{L}^{I2T}(d) + \mathcal{L}^{T2I}(d) \tag{10}$$

Finally, the 3D InfoNCE loss is the summation of all 2D InfoNCE loss.

$$\mathcal{L} = \sum_{d=1}^{3} \mathcal{L}(d) \tag{11}$$

## 3 RESULTS

### 3.1 DATA

We used the MIMIC-CXR training set as the training set, and used Open-I, CheXpert, ChestXray14, and ChestXDet10 as the test sets for zero-shot classification. At the same time, we used the MIMIC-CXR test set as the test set for retrieval-style report generation. Details of the datasets can be found in the appendix.

### 3.2 EVALUATION METRIC

In our experiment, for both zero-shot and fine-tuning classification evaluation metrics, we adapt Area under the ROC Curve (AUC), F1 score (F1), Matthews Correlation Coefficient (MCC), and mean Average Precision (mAP). For F1 score, we followed the evaluation method in CheXzero Tiu et al. (2022) and MedKLIP Wu et al. (2023a), calculating F1 score at the optimal threshold. Therefore, for MCC, we also calculate the scores at the optimal threshold. And for retrieval based report generation evaluation metrics, we adapt common NLG metrics, Recall-Oriented Understudy for Gisting Evaluation (ROUGE) Lin (2004), Bilingual Evaluation Understudy (BLUE) Papineni et al. (2002), and Consensus-based Image Description Evaluation (CIDEr) Vedantam et al. (2015).

### 3.3 IMPLEMENTATION DETAILS

In our experiments, we choose ViT-B/16 Dosovitskiy (2020) as the image encoder which utilizes M3AE Chen et al. (2022) for pretraining on the MIMIC-CXR, and choose BioBERT Lee et al. (2020) as the text encoder which is fine-tuned on MIMIC-CXR, too. Other implementation details can be found in the appendix.

### 3.4 COMPARISON WITH STATE-OF-THE-ART METHODS

**Zero-Shot Classification** We compare the performance of existing SOTA methods in zero-shot classification on four officially released test sets. To ensure a fair comparison, the baseline model's training data excludes any LLM-generated reports. For CARZero Lai et al. (2024), which originally incorporated LLM-generated reports during training, we removed these reports and retrained the model without LLM prompt templates. All other models use their respective officially released parameters for inference. As shown in Table 1, in both the POS and PNC evaluation methods, our model achieves the best performance across all classification metrics on all test datasets. This demonstrates the significant potential of our approach for diagnosing rare diseases and highlights the strong generalization performance of our model in zero-shot classification tasks. It is also worth noting that most models experience varying degrees of performance decline under the PNC evaluation method. This indicates that previous SOTA models can effectively distinguish the semantic similarity between medical images and positive mentions of different disease categories ("inter-class similarity"), but struggle to differentiate the semantic similarity between medical images and both positive and negative mentions within a given disease category ("intra-class similarity"). Our model, however, performs strongly in both evaluation methods, demonstrating its ability to effectively differentiate both "inter-class similarities" and "intra-class similarities", which corresponding to the schematic diagram in Figure 1.

**Fine-Tuning Classification** We compare the performance of existing SOTA methods in fine-tuning classification on 1% Open-I data. As shown in Table 2, our model continues to achieve the best performance on all classification metrics, and even our zero-shot classification performance surpasses

| Method | Evaluation Method | Open-I | | | | CheXpert | | | | ChestXray14 | | | | ChestXDet10 | | | |
|---|---|---|---|---|---|---|---|---|---|---|---|---|---|---|---|---|---|
| | | AUC↑ | F1↑ | MCC↑ | mAP↑ | AUC↑ | F1↑ | MCC↑ | mAP↑ | AUC↑ | F1↑ | MCC↑ | mAP↑ | AUC↑ | F1↑ | MCC↑ | mAP↑ |
| MedCLIP | POS | 0.500 | 0.134 | 0.106 | 0.096 | 0.528 | 0.389 | 0.224 | 0.312 | 0.510 | 0.146 | 0.089 | 0.090 | 0.517 | 0.322 | 0.120 | 0.198 |
| | PNC | 0.756 | 0.184 | 0.190 | 0.118 | 0.819 | 0.531 | 0.455 | 0.452 | 0.704 | 0.180 | 0.158 | 0.110 | 0.647 | 0.347 | 0.222 | 0.260 |
| KAD | POS | 0.818 | 0.283 | 0.279 | 0.228 | 0.849 | 0.549 | 0.471 | 0.532 | 0.796 | 0.289 | 0.259 | 0.221 | 0.749 | 0.449 | 0.470 | 0.392 |
| | PNC | 0.695 | 0.169 | 0.147 | 0.095 | 0.786 | 0.514 | 0.405 | 0.443 | 0.695 | 0.168 | 0.135 | 0.110 | 0.675 | 0.383 | 0.404 | 0.304 |
| CARZero* | POS | 0.802 | 0.192 | 0.273 | 0.212 | 0.857 | 0.227 | 0.487 | 0.545 | 0.761 | 0.146 | 0.213 | 0.164 | 0.732 | 0.203 | 0.343 | 0.381 |
| | PNC | 0.336 | 0.074 | 0.068 | 0.037 | 0.099 | 0.264 | 0.021 | 0.105 | 0.303 | 0.089 | 0.025 | 0.038 | 0.329 | 0.289 | 0.095 | 0.157 |
| VECL(Ours) | POS | **0.829** | **0.345** | **0.341** | **0.277** | **0.900** | **0.649** | **0.579** | **0.655** | **0.815** | **0.305** | **0.275** | **0.277** | **0.800** | **0.493** | **0.421** | **0.470** |
| | PNC | **0.823** | **0.336** | **0.334** | **0.275** | **0.909** | **0.660** | **0.597** | **0.668** | **0.814** | **0.303** | **0.273** | **0.224** | **0.788** | **0.485** | **0.400** | **0.460** |

Table 1: Comparison of different methods on Open-I, CheXpert, ChestXray14, ChestXDet10 for zero-shot classification. To ensure a fair comparison, for CARZero, we removed the LLM-generated data and retrained the model for inference.

the performance of other methods fine-tuned on 1% of the data, proving the strong advantages of our approach.

**Retrieval Based Report Generation** We compare the performance of existing SOTA methods in retrieval based report generation on MIMIC-CXR test set. Also shown in Table 2, our model achieves the best performance on all NLG metrics once again. Interestingly, CARZero, as the last SOTA method, shows the worst performance among several methods in the retrieval based report generation task, while the much earlier method MedCLIP Wang et al. (2022) has demonstrated quite good performance. This may be because, in the retrieval-based report generation task, the model needs not only to distinguish between "inter-class similarities" of different diseases but also to distinguish "intra-class similarities" given a specific disease. The model must clearly determine whether the target disease category exists in order to match the most similar report. By comparing the zero-shot classification metrics of MedCLIP and CARZero in the PNC evaluation method, we can also find that MedCLIP significantly outperforms CARZero, which to some extent supports our explanation. This also reflects that PNC evaluation method can provide a more comprehensive evaluation.

| Method | Open-I | | | | MIMIC-CXR | | | |
|---|---|---|---|---|---|---|---|---|
| | AUC↑ | F1↑ | MCC↑ | mAP↑ | RG-1↑ | BL-1↑ | BL-2↑ | CIDEr↑ |
| MedCLIP | 0.754 | 0.077 | 0.188 | 0.136 | 0.184 | 0.183 | 0.086 | 0.015 |
| KAD | 0.771 | 0.156 | 0.239 | 0.196 | 0.176 | 0.198 | 0.084 | 0.017 |
| CARZero* | 0.815 | 0.183 | 0.285 | 0.219 | 0.150 | 0.180 | 0.079 | 0.014 |
| VECL (Ours) | **0.830** | **0.366** | **0.353** | **0.294** | **0.188** | **0.214** | **0.089** | **0.017** |

Table 2: Comparison of different methods on 1% Open-I data for fine-tuning classification and on MIMIC-CXR test set for retrieval based report generation. To ensure a fair comparison, for CARZero, we removed the LLM-generated data and retrained the model for fine-tuning and report generation.

## 3.5 ABLATION STUDY

**Ablation Study of Visual Entailment** To validate the effectiveness of the visual entailment method in construction of the similarity matrix label, we design control experiment constructing the similarity matrix label without visual entailment method as the baseline. In baseline model, the sample matching relationship represented by three basis vectors degenerates into numbers 0 and 1. And the medical images supporting the report sentences degenerate into positive samples, while the images contradicting or being neutral to the report sentences degenerate into negative samples. As shown in Table 3, We compared the zero-shot classification performance between control experimental group on CheXpert. The baseline model's F1 scores shows significant declines under in evaluation methods, and all metrics in the PNC evaluation method also significantly drop. The degraded baseline model can still model the support and neutral relationships between positive and negative samples, but it fails to model the contradictory relationships of negative mentions. This indicates that modeling the contradictory relationships of negative mentions in contrastive learning is crucial for enhancing the model's ability to distinguish between "inter-class similarities", especially "intra-class similarities".

**Ablation Study of Loss Function** To validate the effectiveness of the 3D InfoNCE loss, we design control experiment replacing the 3D InfoNCE loss with BCE loss and Cross Entropy loss respectively. For BCE loss, we slice the third dimension of the three-dimensional similarity matrix and labels, and then optimize three two-dimensional matrices simultaneously, shape of which is batch size $\times$ batch size ; For Cross Entropy loss, we slice the first two dimensions of the three-dimensional similarity matrix and labels, and then optimize one two-dimensional matrix, shape of which is (batch size·batch size) $\times$ 3. Also shown in Table 3, We compared the zero-shot classification performance between control experimental group on CheXpert. In all control groups, the model using the 3D InfoNCE loss achieved the best performance, demonstrating the effectiveness of the contrastive learning task. Compared to BCE loss and cross-entropy loss, the 3D InfoNCE loss not only considers whether the modeling of the sample itself regarding the relationships of entailment, neutrality, and contradiction is correct, but also requires comparing the entailment, neutrality, and contradiction relationships between samples, increasing the difficulty of the task and thereby enhancing the model's learning ability.

| Visual Entailment | | Loss Function | | | Evaluation Method | CheXpert | | | |
|---|---|---|---|---|---|---|---|---|---|
| False | True | BCE | Cross Entropy | 3D InfoNCE | | AUC↑ | F1↑ | MCC↑ | mAP↑ |
| ✓ | | | | ✓ | POS | 0.899 | 0.331 | 0.574 | 0.639 |
| | | | | | PNC | 0.463 | 0.219 | 0.208 | 0.255 |
| | ✓ | ✓ | | | POS | 0.896 | 0.638 | 0.568 | 0.639 |
| | | | | | PNC | 0.906 | 0.652 | 0.586 | 0.660 |
| | ✓ | | ✓ | | POS | 0.882 | 0.614 | 0.544 | 0.614 |
| | | | | | PNC | 0.893 | 0.623 | 0.552 | 0.639 |
| | ✓ | | | ✓ | POS | **0.900** | **0.649** | **0.579** | **0.655** |
| | | | | | PNC | **0.909** | **0.660** | **0.597** | **0.668** |

Table 3: Ablation study of visual entailment and loss function

**Ablation Study of Label Categories** To explore the influence of the label categories set $C$ on the model performance, we designed control experiments with different label categories sets C. We gradually reduce the number of label categories by intervals of three, and construct new similarity matrices for models training. We compared the zero-shot classification performance among control experimental groups on Open-I. As shown in Figure 7, in both evaluation methods, as the number of label categories gradually decreases, the four classification metrics overall show a downward trend, indicating that the number of label categories can affect the model's performance, and belong to negative correlation. The richer the label categories, the more the model can use visual entailment method to model samples relationships, providing richer supervisory signals, which helps the model capture more complex semantic relationships among samples. Looking at the absolute values of the metrics, although the classification metric scores overall decrease as the number of label categories gradually decreases, the decline is not significant and are still higher than almost all other methods. This suggests that our method has a low dependency on the number of label categories and possesses a higher degree of robustness.

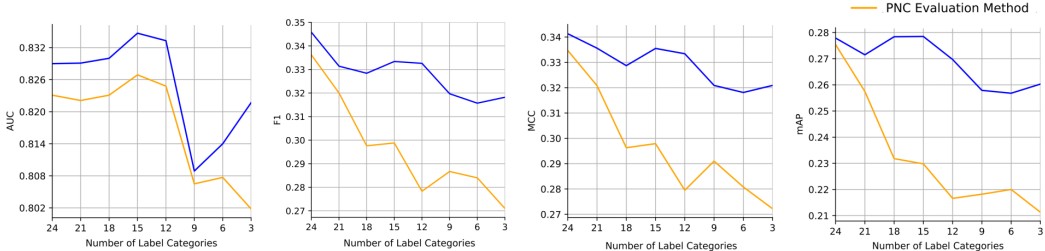

Figure 7: The ablation study of label categories. We gradually reduce the number of label categories by intervals of three, and comparing the zero-shot classification performance in different evaluation method. As the number of label categories gradually decreases, the decline of metric scores is not significant, suggesting that our method has a low dependency on the number of label categories and possesses a higher degree of robustness

### 3.6 VISUALIZATION

In order to further verify our analysis in Figure 3, we selected CARZero as the baseline for comparison with our method. We performed t-SNE visualization on the similarities output from the model's fusion module. We set up two control experiments, one comparing the distributions of positive images + positive text fusion features with positive images + negative text fusion features under the given disease category, and the other comparing the distributions of negative images + positive text fusion features with negative images + negative text fusion features the given disease category. As shown in Figure 8, In each specified disease category, our method can effectively distinguish between the distributions of the two types of features, while CARZero is much poorer. Specifically, the results above the horizontal line represent the comparison of the distributions between positive images + positive text fusion features and positive images + negative text fusion features under a given disease category. The results below the horizontal line represent the comparison of the distributions between negative images + positive text fusion features and negative images + negative text fusion features under a given disease category. This indicates that our model can bring positive images closer to positive text and farther from negative text, while simultaneously bringing negative images closer to negative text and farther from positive text in the feature space, which corresponding to the schematic diagram in Figure 3. CARZero fails to do so, leading to a significant decline in all metrics in PNC evaluation method on the zero-shot classification task.

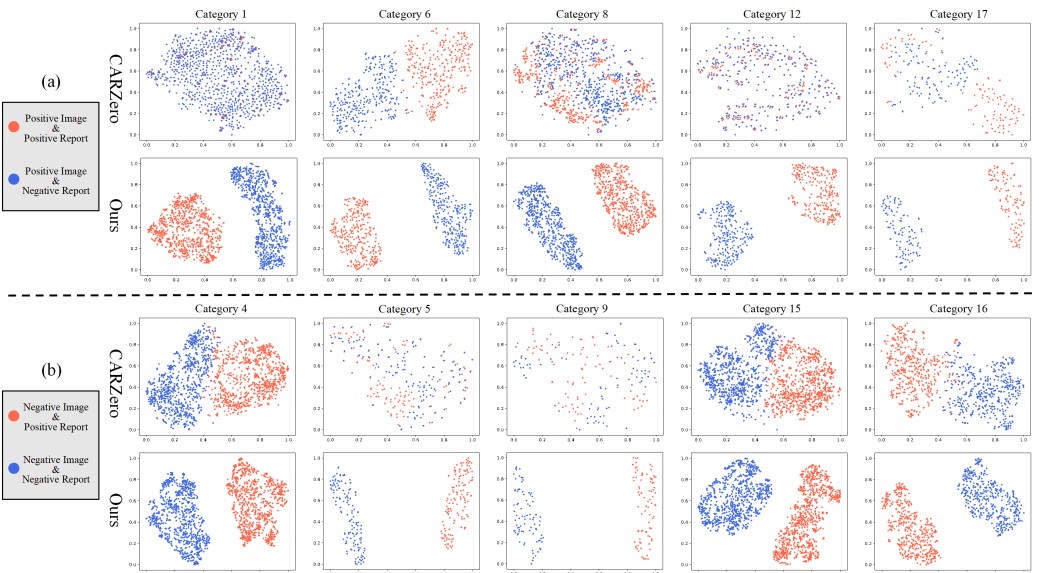

Figure 8: t-SNE visualization on the similarities output from the model's fusion module between CARZero and our VECL. We compared the distributions of positive images + positive text fusion features with positive images + negative text fusion features under the given disease category, as well as the distributions of negative images + positive text fusion features with negative images + negative text fusion features the given disease category. Visualization results indicate that VECL can better distinguish between positive and negative mentions, which corresponding to the schematic diagram in Figure 3

## 4 RELATED WORK

### 4.1 CONTRASTIVE LEARNING IN VISION-LANGUAGE PRETRAINING

Our approach builds upon contrastive learning-based vision-language pretraining methods, which have achieved notable success in both general and medical domains. In the general domain, CLIP Radford et al. (2021) has set a benchmark for joint visual-textual representation learning using large-scale image-text pairs.

In the medical domain, several methods have adapted this paradigm for radiology data. ConVIRT Zhang et al. (2022) pioneered the use of contrastive learning to align medical scans with their cor-

responding reports. GLoRIA Huang et al. (2021) extends this with fine-grained alignment between global and local features of medical images and text descriptions. MedKLIP Wu et al. (2023a) incorporates prior knowledge through disease descriptions to enhance representation learning. KAD Zhang et al. (2023) utilizes entity extraction from image-associated reports, combined with their semantic types, to perform contrastive learning via a knowledge encoder. These methods align modalities primarily through cosine similarity.

CARZero Lai et al. (2024) introduces cross-attention alignment to capture the nuanced relationships between medical images and reports. However, it overlooks detailed report information, resulting in suboptimal feature representations and difficulties in distinguishing between positive and negative mentions. MedCLIP Wang et al. (2022) addresses this issue by using entity extraction tools to convert report sentences into multi-hot vectors, applying a soft semantic matching loss, while still relying on cosine similarity.

In contrast, our approach utilizes a large language model (LLM) to extract both positive and negative mentions of diseases from reports. Through a cross-attention mechanism within a visual entailment framework, we optimize the model to learn more robust and fine-grained representations, improving its ability to distinguish between nuanced medical conditions.

### 4.2 VISUAL ENTAILMENT

Visual entailment aims to determine the relationship between a premise and a hypothesis, classifying it as entailment, neutral, or contradiction. Unlike NLI tasks MacCartney (2009), where the premise is textual, visual entailment uses images as premises. The SNLI-VE dataset Xie et al. (2019), the most commonly used dataset for this task, is adapted from SNLI Bowman et al. (2015), replacing textual premises with images from Flickr30k Young et al. (2014).

Traditional visual entailment models use classification-based frameworks to directly predict one of the three relationships between an image and a hypothesis. In contrast, we incorporate these relationships into a contrastive learning framework by introducing an extended version of the InfoNCE loss Oord et al. (2018).

## 5 CONCLUSION

In this paper, we suggest adding a evaluation method for medical vision-language models, Positive-Negative Contrastive (PNC) evaluation method, and a Visual Entailment-based Contrastive Learning (VECL) approach, emphasizing the importance of considering positive and negative mentions in medical image-text pairs. Experiments demonstrate that integrating PNC evaluation method provides a more comprehensive assessment of model performance, while VECL achieves state-of-the-art results across various downstream tasks. Ablation studies validate the effectiveness of the visual entailment method in constructing similarity matrix labels and explore the impact of label category sets on model performance. Finally, t-SNE visualization reveals the reasons why VECL achieves the best performance in PNC evaluation method. We hope this paper provides new perspectives for vision-language models in the medical domain, potentially benefiting future research and clinical practice.

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

## A APPENDIX

### A.1 DATASET

**MIMIC-CXR** Johnson et al. (2019) The MIMIC Chest X-ray (MIMIC-CXR) Database is a large publicly available dataset of chest radiographs in DICOM format with free-text radiology reports. The dataset comprises 377,110 images corresponding to 227,835 radiographic studies conducted on 65,379 patients. Each radiographic study is accompanied by a radiology report and the corresponding chest X-ray image, which may be frontal or lateral views. The radiology report serves as a comprehensive summary provided by radiologists, encompassing various sections such as examination, indication, impression, findings, technique, and comparison. In this study, we use MIMIC-CXR training set for training and MIMIC-CXR test set for retrieval based report generation evaluation. After data cleaning, the training set, test set, and validation set each contain 228,594, 3,858, and 3,018 image-text pairs, respectively. For image data, we select the frontal views of chest X-ray image, and for text data, we select the findings and impressions sections.

**Open-I** Demner-Fushman et al. (2016) Open-I contains 3,955 reports and 7,470 Chest X-ray images, which includes manual annotations for 18 different multi-label diseases. In this study, we use Open-I for zero-shot and finetune classification evaluation.

**CheXpert** Irvin et al. (2019) CheXpert has 224,316 CXRs collected from 65,240 patients. The official test set contains 500 patients annotated by a consensus of 5 board-certified radiologists: Atelectasis, Cardiomegaly, Consolidation, Edema, and Pleural Effusion. In this study, we use the official test set and only the above five disease labels for zero-shot classification evaluation.

**ChestXray14** Wang et al. (2017) NIH ChestXray14 has 112,120 chest X-ray images with 14 disease labels from 30,805 unique patients. The official test set released by the NIH, comprising 22,433 images, are distinctively annotated by boardcertified radiologists. In this study, we use the official test set for zero-shot classification evaluation.

**ChestXDet10** Liu et al. (2020) ChestX-Det10 is a subset of NIH ChestXray14, which is consisting of 3543 CXRs with boxlevel annotations provided by 3 board-certified radiologists of 10 diseases. The official test set contains 542 CXRs with 10 diseases and corresponding box-level annotations. In this study, we use the official test set for zero-shot classification evaluation.

## A.2  IMPLEMENTATION DETAILS

The fusion module employs shared weights for both 'I2T' and 'T2I' alignments. Images are standardized to a size of $224 \times 224$ pixels. We implement standard data augmentation techniques such as random horizontal flips, random affine transformations, and color jittering. After segmented into sentences by LLM, a random sentence capped at 97 characters selected per training cycle. The LLM we use is Meta-Llama-3-8B-Instruct AI@Meta (2024). The Adam optimizer is utilized with a learning rate of 5e-5. All experiments are conducted with an 80G A800 GPU.

## A.3  LLM LABLE CATEGORY

In order from top to bottom, each category corresponds to 1, 2, ..., 24 in the LLM Label set $C$, and Others corresponds to 25.

| Train | Test | | | |
|---|---|---|---|---|
| MIMIC | OpenI | ChestXDet10 | ChestXray14 | Chexpert |
| Atelectasis | Atelectasis | Atelectasis | Atelectasis | Atelectasis |
| Pleural Effusion | Pleural Effusion | Pleural Effusion | Pleural Effusion | Pleural Effusion |
| Pneumothorax | Pneumothorax | Pneumothorax | Pneumothorax | |
| Cardiomegaly | Cardiomegaly | | Cardiomegaly | Cardiomegaly |
| Lung Opacity | Lung Opacity | | | |
| Pneumonia | Pneumonia | | Pneumonia | |
| | Pulmonary Mass | Pulmonary Mass | Pulmonary Mass | |
| Edema | Pulmonary Edema | | Pulmonary Edema | Pulmonary Edema |
| | Lung Nodule | Lung Nodule | Lung Nodule | |
| | Lung Infiltration | | Pulmonary Infiltration | |
| | Pulmonary Fibrosis | Fibrosis | Fibrosis | |
| | Pulmonary Emphysema | Pulmonary Emphysema | Pulmonary Emphisema | |
| | Pleural Thickening | | Pleural Thickening | |
| | Hernia | | Hernia | |
| Consolidation | | Pulmonary Consolidation | Pulmonary Consolidation | Pulmonary Consolidation |
| Fracture | Bone Fracture | Bone Fracture | | |
| Enlarged Cardiomediastinum | | | | |
| Pleural Other | | | | |
| Lung Lesion | | | | |
| Support Devices | | | | |
| | Abnormal Lesion | | | |
| | Lung Granuloma | | | |
| | Calcified Granuloma | | | |
| | | Tissue Calcification | | |
| Others | Others | Others | Others | Others |

Table 4: LLM Lable Category

## A.4  LLM PROMPT

An example of the prompt for tagging using LLM is as follows, and detailed prompts are available in the code repository.

## A AN EXAMPLE OF THE LLM PROMPT

```
def prompt_data(sent):
    prompt = """
    Assume you are an experienced radiologist. Help me identify the
        correct medical condition label for the given radiology report
        sentence. Below are the medical condition labels with
        corresponding numbers and their medical descriptions:

1.Atelectasis: Lung tissue exhibits signs of partial or complete
    atelectasis, with decreased lung volume and increased localized
    radiographic density.
2.Pleural Effusion: There is an abnormal accumulation of fluid within the
     pleural cavity, which is evident on X-ray imaging as a blunted
    costophrenic angle or the presence of a fluid level.
3.Pneumothorax: There is evidence of free air within the thoracic cavity,
     resulting in partial or complete atelectasis. This is characterized
    by a retracted lung edge and an area devoid of lung markings.
4.Cardiomegaly: The cardiac silhouette is enlarged, indicating a size
    beyond the normal range.
5.Opacity: A lung region demonstrates increased radiopacity, potentially
    suggestive of inflammatory changes, a tumor, or hemorrhage.
6.Pneumonia: Patchy to diffuse lung infiltrates are observed, frequently
    associated with air bronchograms.
7.Pulmonary Mass: A lung mass, either well-circumscribed or poorly
    defined, is present, typically measuring over 3 cm in diameter.
8.Edema: Interstitial or alveolar fluid accumulation in the lungs is
    evident, characterized by increased and indistinct lung markings, a
    common finding in cardiogenic pulmonary edema.
9.Lung Nodule: A round or oval opacity within the lung, measuring less
    than 3 cm in diameter.
10.Lung Infiltration: Patchy or reticular opacities are noted within the
    lung tissue, suggestive of inflammatory processes or other
    infiltrative conditions.
11.Fibrosis: Interstitial lung thickening and fibrosis are present,
    exhibiting a reticular pattern and a honeycomb-like appearance on
    imaging studies.
12.Emphysema: Overinflation of the lungs with alveolar destruction is
    observed, manifesting as increased lung lucency and expanded lung
    fields on the chest X-ray.
13.Pleural Thickening: Pleural layer thickening is evident, appearing as
    broadened pleural shadows on imaging, often attributed to chronic
    inflammation or fibrosis.
14.Hernia: Internal organ protrusion through either normal or abnormal
    openings is observed; in the case of a diaphragmatic hernia, this may
     present as an abnormal position and contour of the diaphragm on X-
    ray imaging.
15.Consolidation: The lung alveoli are opacified with fluid or solid
    material, manifesting as regions of increased density with indistinct
     borders on imaging. the lung tissue exhibits a solidified appearance
    , a common finding in cases of pneumonia.
16.Bone Fracture: A discontinuity within the bone structure is observed
    on X-ray, characterized by a disrupted cortical bone and the presence
     of fracture lines.
17.Enlarged Cardiomediastinum: An enlargement of the mediastinal shadow
    or cardiomediastinal silhouette is noted.
18.Pleural Other: Other pleural abnormalities, such as pleural
    calcifications or plaques, are present, exhibiting specific imaging
    features that are indicative of the underlying condition.
19.Lung Lesion: An encompassing term for a variety of abnormal imaging
    findings within the lung, encompassing nodules, masses, infiltrates,
    and other anomalies.
20.Support Devices: Visualized on imaging are various medical devices,
    including catheters, stents, prosthetic heart valves, and other
    implanted or inserted medical apparatus.
```

```
21.Abnormal Lesion: A non-specific term denoting any abnormal imaging
     findings within the lungs, which may encompass a range of
     presentations such as nodules, masses, opacities, or other anomalies.
22.Lung Granuloma: A small pulmonary nodule, usually measuring less than
     3 cm in diameter, frequently exhibiting calcification.
23.Calcified Granuloma: A calcified granuloma is observed, manifesting as
     a high-density nodule on imaging studies.
24.Tissue Calcification: Calcifications within soft tissue are noted,
     appearing as areas of increased density on imaging, indicative of
     calcified spots or plaques.
25.No Mention: None of the above symptoms are mentioned or related,
     cannot exist with any other labels at the same time.

Examples:

Sentence: there is no focal consolidation pleural effusion or
     pneumothorax.
Label: 15, 2, 3

Sentence: bilateral nodular opacities that most likely represent nipple
     shadows.
Label: 9

Sentence: chronic deformity of the posterior left sixth and seventh ribs
     are noted.
Label: 16

Sentence: the patient shows no signs of free air below the right
     hemidiaphragm.
Label: 3

Sentence: the imaged upper abdomen shows no remarkable findings.
Label: 25

Sentence: the patient's overall condition is normal.
Label: 25

Special Note:
If the sentence mentions a condition (whether positive or negative), use
     the corresponding label.
If the sentence describes multiple conditions, except 25(No Mention),
     output multiple labels, separated by commas ",".
Remember, 25(No Mention) and other labels cannot exist at the same time.

In case you forgot, let me repeat these labels:
1. Atelectasis
2. Pleural Effusion
3. Pneumothorax
4. Cardiomegaly
5. Opacity
6. Pneumonia
7. Pulmonary Mass
8. Edema
9. Lung Nodule
10. Lung Infiltration
11. Fibrosis
12. Emphysema
13. Pleural Thickening
14. Hernia
15. Consolidation
16. Bone Fracture
17. Enlarged Cardiomediastinum
18. Pleural Other
19. Lung Lesion
20. Support Devices
```

```
77   21. Abnormal Lesion
78   22. Lung Granuloma
79   23. Calcified Granuloma
80   24. Tissue Calcification
81   25. No Mention
82
83   Now, please select the most appropriate label for the following sentence
          and output only the corresponding number(s).
84   Notice: you only need to output the label (pure numbers), do not output
          anything else!
85   """
86       messages = [
87           {"role": "system", "content": prompt},
88           {"role": "user", "content": sent}
89       ]
90       return messages
```

