# OpenReview forum: "Medical Vision-Language Pretraining through Contrastive Learning of Positive and Negative Mention"
_ICLR.cc/2025/Conference — ICLR 2025 Conference Withdrawn Submission_

### Official Review · Reviewer_3RKJ · 2024-10-30

**Soundness:** 3
**Presentation:** 3
**Contribution:** 2
**Rating:** 5
**Confidence:** 5

**Summary:**

This paper focuses on medical vision language pre-training using chest X-ray datasets. The proposed contrastive approach explicitly models entailment, contradiction, and neutral relationships between medical images and report sentences. Experiments on various downstream tasks demonstrate the effectiveness.

**Strengths:**

This paper leverages a special characteristic of medical reports - the presence of both positive and negative mentions - and proposes a novel solution through Visual Entailment-based Contrastive Learning (VECL). The effectiveness of this approach is well demonstrated through t-SNE visualizations, which clearly show the model's superior ability to distinguish between positive and negative mentions in the feature space compared to existing methods. The writing is well-structured and easy to follow.

**Weaknesses:**

1. The claim of PNC evaluation as a novel contribution is questionable, as similar evaluation methods have been previously published [1]. "We define the procedure as follows. First, we compute logits with positive prompts (such as atelectasis) and negative prompts (that is, no atelectasis). Then, we compute the softmax between the positive and negative logits. Lastly, we keep the softmax probabilities of the positive logits as the probability that the disease is present in the chest X-ray." This is similar to the definition of PNC in the paper "in PNC evaluation method, both positive and negative mentions of different disease categories are used as text inputs, designated as “There is {disease}” and “There is no {disease}”. After calculating the similarity between all texts and images, the similarities of positive and negative mentions within the same category are normalized by softmax, and then the normalized."
2. Evaluating models trained only with POS using PNC metrics is unfair. During training, POS models are only exposed to positive mention patterns (e.g., "disease"), so the models may fail when processing negative mentions (e.g., "no disease") during evaluation. The model's text encoder, having never learned to distinguish the semantic significance of negation words like "no," may primarily focus on the disease term while failing to capture the negation context. This could result in similar embeddings for both "disease" and "no disease" prompts, undermining the validity of the PNC evaluation metrics. (Minor, Figure 2 PC should be POS?).
3. Figure 7 shows faster performance degradation in PNC compared to POS as categories decrease, but lacks theoretical explanation.
4. Some recent work[2]  reports significantly higher performance on similar tasks.
5. The visualization analysis effectively demonstrates the model's advantages, however, it only compares with CARZero - including more baseline models would strengthen the empirical validation.

[1] Tiu, E., Talius, E., Patel, P. et al. Expert-level detection of pathologies from unannotated chest X-ray images via self-supervised learning. Nat. Biomed. Eng (2022). https://doi.org/10.1038/s41551-022-00936-9.
[2] Dai, Tianjie, et al. "UniChest: Conquer-and-Divide Pre-training for Multi-Source Chest X-Ray Classification." IEEE Transactions on Medical Imaging (2024).

**Questions:**

1. Figure 7 shows that the number of label categories gradually decreases, the four classification metrics overall show a downward trend. Why only use 24 categories? Dataset like PadChest has a much larger label set.

2. What's the problem with the Positive-Only Similarity (POS) evaluation method in the real-clinical scenario?

---

### Official Review · Reviewer_DxB8 · 2024-11-01

**Soundness:** 3
**Presentation:** 3
**Contribution:** 2
**Rating:** 5
**Confidence:** 4

**Summary:**

This paper addresses a limitation in current vision-language pre-training models for the medical domain, particularly in their handling of negative mentions in radiology reports. The authors introduce a Positive-Negative Contrastive (PNC) evaluation method to assess the models' capacity to distinguish between positive and negative mentions, thereby providing a more comprehensive evaluation framework. Additionally, they propose a Visual Entailment-based Contrastive Learning (VECL) method, which extends traditional contrastive learning by incorporating entailment, contradiction, and neutrality relationships. This approach adapts the InfoNCE loss function to a three-dimensional structure, enabling more nuanced alignment between medical images and report sentences. Experimental results show that these proposed methods enhance model performance in downstream tasks, demonstrating state-of-the-art results across multiple metrics.

**Strengths:**

- The VECL method introduces a novel approach, addressing complex semantic relationships within medical data by leveraging Visual Entailment to model entailment, contradiction, and neutrality relationships.
- They use the PNC evaluation method to provide a more realistic assessment framework by capturing nuanced positive and negative mentions within radiology reports
- Experimental results indicate that the proposed methods significantly improve model performance across various downstream tasks.

**Weaknesses:**

- Constructing visual entailment relationships depends on LLMs for disease category extraction, making quality control challenging, especially when using models like Meta-Llama-3-8B-Instruct for tagging.
- The PNC evaluation method has already been utilized in previous works, including one of this study's baselines, CheXzero. The primary difference is in the prompt format (“{Pathology}”, “No {Pathology}” → “There is {disease}” and “There is no {disease}”).
[1] Expert-level detection of pathologies from unannotated chest X-ray images via self-supervised learning, Tiu et al., Nature Biomedical Engineering, 6(12):1399–1406, 2022.
[2] Xplainer: From X-Ray Observations to Explainable Zero-Shot Diagnosis, Pellegrini et al., MICCAI 2023.
[3] Significantly improving zero-shot X-ray pathology classification via fine-tuning pre-trained image-text encoders, Jang et al., Sci Rep 14, 23199 (2024).
- This model uses 25 labels extracted from the test reports for training, covering all labels in their test set. Such a setup makes zero-shot classification comparisons unfair. A more fair comparison would involve an out-domain dataset and labels, such as PadChest.

**Questions:**

- Why not use PadChest and PadChest20 following the CARZero evaluation setting?
- You employ the CARZero-pretrained vision encoder, but then compare it with a CARZero variant that you re-trained without the LLM-augmented model, which may create an unfair comparison.
- CARZero uses both original and LLM-augmented reports paired with a single CXR image, while your model uses sentence-level text paired with the corresponding CXR image. It would be more accurate to compare with the original CARZero model provided by the authors.

---

### Official Review · Reviewer_RBxm · 2024-11-02

**Soundness:** 2
**Presentation:** 3
**Contribution:** 2
**Rating:** 5
**Confidence:** 4

**Summary:**

This paper proposes an improved vision-language pretraining approach that tailors evaluation to the unique characteristics of medical data, specifically the presence of both positive and negative disease mentions in radiology reports. To better reflect this, the authors introduce a zero-shot classification evaluation that accounts for these distinctions, providing a more accurate assessment. They also present a contrastive learning method that models entailment, contradiction, and neutral relationships between images and report sentences, achieving state-of-the-art results and enhancing model comprehension of nuanced medical image-report relationships.

**Strengths:**

The strengths of this paper include:
1. The paper identifies a critical distinction in medical data, where radiology reports contain both positive and negative disease mentions, unlike general image-text pairs. By incorporating this unique feature, the paper offers a more nuanced approach to evaluating model performance in the medical domain.
2. The introduction of the PNC evaluation method allows for a more comprehensive evaluation by considering both positive and negative mentions, thus better reflecting real-world medical scenarios and revealing model limitations that were previously overlooked with Positive-Only Similarity (POS) evaluation.
3. The Visual Entailment based Contrastive Learning (VECL) method extends traditional contrastive learning by modeling entailment, contradiction, and neutral relationships between images and report sentences. This approach captures complex semantic relationships, which are especially important in the medical domain.
4. The experimental result shows the effectiveness of PNC evaluation and VECL through experiments, revealing deficiencies in existing models and achieving improved performance in downstream tasks.

**Weaknesses:**

The paper lacks sufficient clarity and justification in key areas. Definitions and grouping of labels, particularly the neutral classification and the merging of distinct categories like ‘other diseases’ and ‘normal,’ are not well-explained, raising questions about label consistency. The evaluation relies too heavily on NLG metrics, omitting a clinically relevant, label-based comparison to verify accurate report matching. Claims about the model’s capability in handling rare diseases lack substantiating performance metrics. Details regarding data cleaning methods and the accuracy of LLM-based label extraction remain unverified, and certain experimental aspects, such as selection criteria in ablation studies, are insufficiently described. For further details, please refer to the Questions section.

**Questions:**

- Q1: In lines 212 to 213, it is stated that ‘If the image label set neither includes the text label nor its opposite label, we consider the relationship between the image and the report sentence to be neutral.’ Given that the image label set is derived from the complete report and that the text label is one sampled label from the report sentence (‘only one label of the sampled report sentence as text label and all labels of a complete report as the image label set’), it seems the image label set would inherently include either the text label or its opposite. It would be helpful if the authors could provide specific examples or scenarios where the neutral case might occur, if it is indeed possible. Clarifying this would ensure that this scenario is either a valid part of the methodology or highlight if further adjustments to the description are needed.
- Q2: There is a question regarding label category 25, where ‘other diseases’ and ‘normal’ are grouped together despite being fundamentally different categories. The rationale for this grouping appears insufficiently explained, and further clarification would be beneficial, particularly since this choice could significantly impact the model’s performance and interpretability. It would be helpful if the authors could provide their reasoning for grouping these categories and discuss the implications on the model’s ability to distinguish between normal cases and other diseases. Additionally, examples illustrating how category 25 is handled in practice would offer valuable insight into the model’s decision-making process.
- Q3: In the retrieval-based report generation task, simply comparing performance using NLG metrics is insufficient. The paper has already established a framework for extracting labels from reports using LLMs. It would be beneficial to extract labels from both the ground truth reports and the retrieved reports, and then compare these labels to evaluate whether the retrieved reports accurately correspond to the ground truth. Relying solely on NLG results does not adequately assess whether similar reports are being matched correctly.
- Q4: In line 312, it is stated, ‘This demonstrates the significant potential of our approach for diagnosing rare diseases.’ However, clarification is needed regarding the basis for this claim. If the performance on the categories with the least number of samples in each dataset can be shown as evidence, it would be helpful to include that. The proposed method may struggle with rare diseases, which typically represent a small portion of the dataset. It is unclear whether the current approach can effectively handle long-tailed distributions. Is there a direction for future work that addresses this issue?
- Q5: In line 646, it is mentioned that data cleaning was performed on the mimic-cxr dataset, but there is no explanation provided for this process. It would be helpful to include details on the methods and procedures used for data cleaning.
- Q6: It is stated that labels are extracted from each sentence using LLMs. Has the accuracy of this extraction been verified? In the appendix, there are datasets that contain labels annotated directly by radiologists. It would be beneficial to compare these annotated labels with those extracted by the LLM to assess the accuracy of the extraction process.
- Q7: Additionally, it would be important to check whether the number of labels in each dataset aligns with the number of labels extracted by the LLM. A comparison of these quantities would help assess the consistency and reliability of the label extraction process.
- Q8:  In the ablation studies regarding label categories, it is mentioned that ‘We gradually reduce the number of label categories by intervals of three.’ Could you clarify whether the three categories that are excluded are chosen randomly? Providing more detail on the selection process would enhance the understanding of this aspect of the study.
- Q9: In section 3.6 on Visualization, it is mentioned that CARZero is used. Can you confirm whether CARZero excludes any LLM-generated reports, consistent with the previous experiments? Clarifying this would help in understanding the integrity of the visualization results.
- Q10: To further substantiate the claims regarding the model’s capabilities in differentiating both ‘inter-class similarities’ and ‘intra-class similarities,’ it would be beneficial to conduct additional experiments using a Negative-Only Similarity (NOS) evaluation method. Introduced here as a complement to the Positive-Only Similarity (POS) method discussed in the paper, the NOS method focuses exclusively on evaluating the model’s performance in distinguishing between negative samples and their corresponding textual labels. This approach would provide a clearer understanding of how well the model handles ‘purely negative’ cases, enhancing the assessment of the model’s overall effectiveness and robustness in real-world scenarios.

---

### Official Review · Reviewer_aYhL · 2024-11-04

**Soundness:** 1
**Presentation:** 2
**Contribution:** 1
**Rating:** 3
**Confidence:** 5

**Summary:**

The paper presents an approach to medical vision-language pre-training by introducing a visual entailment-based contrastive learning method. This technique aims to enhance the understanding of complex semantic relationships between medical images and radiology reports by explicitly modeling entailment, contradiction, and neutrality among them.

**Strengths:**

The paper is well-written and easy to follow. Generally, the figures are clear.

**Weaknesses:**

My main concerns fall in the nolvelty.

1. The proposed Positive-Negative Contrastive (PNC) Evaluation Method lacks significance for medical vision-language models. In the case of CheXZero (https://www.nature.com/articles/s41551-022-00936-9), the authors have employed positive-negative pair prompts, a method established in 2022.

Furthermore, I believe this distinction is not crucial for medical VLMs. The choice between using positive-only prompts or positive-negative pairs is contingent upon the training pipeline and recommendations outlined in the original papers. It is entirely reasonable to compare these methods directly, as they all provide a measure of similarity to assess the likelihood of a specific disease and no difference between them in practical use. The introduction of PNC evaluation settings does not offer any additional advantages. Therefore, I fail to see the necessity of introducing the PNC framework.

2. The VECL is also not interesting enough. The textual pre-process methods are similar to MedKLIP (CVPR 2023 https://openaccess.thecvf.com/content/ICCV2023/papers/Wu_MedKLIP_Medical_Knowledge_Enhanced_Language-Image_Pre-Training_for_X-ray_Diagnosis_ICCV_2023_paper.pdf) which used triplets supervision signals to maintain and emphasize entailment, contradiction, and neutral relationships.

Based on the above concerns I don't feel that this paper is attractive enough to ICLR attendees to reach the publication requirements.

**Questions:**

For details, I have no more questions. Generally, I believe the details are clearly stated but, as I mentioned in weakness, the key contributions are not significant or interesting enough.

---

> ### Author Response · Authors · 2024-11-14
>
> 1. About PNC
> In actual medical scenarios, models need not only to distinguish between inter-class similarities (corresponding to different disease categories) but also to differentiate intra-class similarities (corresponding to negative and positive descriptions within the same disease category). In medical image-text paired data, the text contains a large number of negative mention (i.e., no something). For example, we have analyzed the MIMIC-CXR test set, where samples containing negative mention account for 49%, which is a significant difference from general image-text paired data. If the model cannot effectively distinguish between positive and negative descriptions, it will have deficiencies when processing actual medical image-text pairs. The traditional POS metric can only evaluate the model's ability to distinguish between inter-class similarities, while the PNC metric also has the capability to assess intra-class similarities, which is very important in actual medical scenarios. For instance, given an image containing pneumonia and two texts, "there is pneumonia" and "there is pleural effusion," the POS metric is higher when the model calculates a higher similarity with the text "there is pneumonia." If the texts are replaced with "there is pneumonia" and "there is no pneumonia," the POS metric remains unchanged regardless of which text the model calculates a higher similarity with; this distinction can only be reflected in the PNC metric.
>
> Regarding the PNC Evaluation Method, it seems that my paper may not have been articulated clearly enough. We are indeed not the first to propose this metric, so we have revised our paper's contribution to suggest a renewed emphasis on the PNC metric. CheXZero was the first to use the PNC metric, but it is a medical field paper and did not have much impact in the computer community. The mainstream work that followed still primarily used the POS metric. Therefore, our main contribution lies in suggesting a re-emphasiz on this metric. Through experimental results, we have analyzed the function of PNC metric evaluation, supplemented the missing evaluation parts of POS metric evaluation, and highlighted the importance of the PNC metric, which is a contribution we have completed for the first time.
>
> 2. About VECL
> I don't think MedKLIP has a significant relationship with our method. MedKLIP mainly adopts a triplet-based approach for knowledge enhancement, where entailment, contradiction, and neutral relationships are just part of its triplets. Its method also includes entities and locations as two other elements, and the embedding approach、the training approach for the constructed triplets is quite different from the similarity label matrix we proposed. More importantly, even though we both involve the concept of visual entailment, our method significantly outperforms MedKLIP in all metrics, including Zero-Shot Classification, Fine-Tuning Classification, Retrieval Based Report Generation, and Grouding in four experiments. I strongly disagree with the reviewer's opinion. Does it mean that if the design ideas are similar, the significant performance improvement by the subsequent work loses all innovative significance? Besides, our design idea only shares a common point with MedKLIP in the aspect of visual entailment, and there is no relation in other parts.

---

### Official Review · Reviewer_9YPG · 2024-11-04

**Soundness:** 3
**Presentation:** 3
**Contribution:** 3
**Rating:** 5
**Confidence:** 4

**Summary:**

This paper propose a new evaluation method, Positive-Negative Contrastie(PNC), that can differenciate the postive and negative mention. The authors argue that this is important since there are many negative mentioning in medical report. And this method can extend the exisiting contrastive learning based model, such as CLIP, to medical domain. Furthermore, the authors introduce a Visual Entailment Based Contrastive Learning(VECL) method to increase the Contrastive models performance on both PNC and POS tasks.

This paper indeed target on an important topic in Contrastive learning, especially in medical domain. But there are some confusion remain to  be explained, please refer to the Weakness part. If the authors give me a decent response I would consider rasie my rating.

**Strengths:**

1. This paper is dedicating on an important topic, negative sampling, in contrastive learning. And showing the limitations of current SOTA works on this PNC tasks.
2. The 3DinfoNCE loss design is interesting and definitely solve the problem of how to include both positive and negative labels into the contrastive learning process.
3. The experiments result seems good, if these results are solid. This model can further become a foundation work for many work and application in medical Vision-Language Models research topics.

**Weaknesses:**

1. This work first claim that negative mentioning is a distinctive difference between medical and general image domain, however, I didn't see any evidence to support this claim. Though I partially approved this claim with some emperical bias and the experiments results also reflects this pattern to some extend. I still want to see a clear statistic comparison between medical and general image domain about your claim.
2. I really want to suggest the authors to reformat or reporduce their figures, especially for Figure 2, I can barely see the '-' negative sign in Fig2(b). I have to enlarge the figures to 150% on my 27 inch monitor to see that. Other reviewers or readers might neglect such important signs with a glimpse. As your major improvement here is all about negative samples, why not make it obvious and highlighted?
3. This work relies on LLMs to extract positive and negative labels but not showing any evaluation on the LLMs performance on this task. Is the LLM employed and the labels it generate really reliable, at what degree of certainity? This is crucial for your further exploration.
4. For equation (2), as you described, you only sample one sentence from a report j, and you use $y_j$ as the text input. This confuse me in following ways: a. would you re-sample a sentence from the same report later in your training process? b. since you are randomly sampling a sentence $k^m$ from a report j on epoch m, I would assume that for next epoch, the sampled sentence $k^{m+1}$ is different sentence from last epoch. Then, the input $y_j$ is actually different now. Are you supposed to use $y_{jk}$ to represent the randomly selected sentence k from report j in your equation?
5. Image-Text retrieval is one of the most important use of CLIP model, but you only report the retrieval-based generation task. I'd like to see some experiments results on image-text retrieval tasks on the MIMIC-CXR dataset. And I think this might be one of the most important contribution of the this work to the community, as we don't have a very efficient tool for medical image-text retrieval models on the market.

**Questions:**

1. In my opinion, introducing negative samples are quite crucial for contrastive learning and can make the model less data-hungery, which means the model could be trained more efficiently with limited data. However, not all negative samples are beneficial for the batch-level trianing. There are some previous research paper suggest that introducing certain sampling methods can increase the negtive-sampling efficiency. I wonder whether this work consider this direction and do some experiments on it, if so, I really appreciate releasing some results on this phenomenon, even just some intermediate results would help.
2. Could you elaborate more on eq(5) - eq(7), why there is a row-wise $L_0^{I2T}$ and column-wise $L_1^{I2T}$ seperately?
3. What is the negative
4. Need more detial for Retrieval Based Report Generation

---

### Official Review · Reviewer_jURR · 2024-11-04

**Soundness:** 2
**Presentation:** 2
**Contribution:** 2
**Rating:** 3
**Confidence:** 4

**Summary:**

**Summary**: This paper improves medical vision-language pre-training via two aspects:
  - (i) Evaluation: Existing studies only assess the semantic similarity between medical images and positive mentions of diseases while this study proposes to normalize the semantic similarity with the negative mentions.
  - (ii) Methodology: This study introduce a visual entailment based contrastive learning method to model the relationships between images and reports.

Experimental results show that the evaluation provides a more comprehensive assessment of performance and the proposed approach achieves state-of-the-art results across various downstream tasks.

**Strengths:**

* **Topic**: Overall, finding the true positive samples and true (hard) negative samples is an important direction to learn better visual representation.
* **Writing**: The paper is well-written and easy to follow.
* **Code**: The code is attached and well-documented.

**Weaknesses:**

* **Motivation**: The motivation "previous studies only evaluate the distance between both negative and positive mentions" is not correct. Studies like [1], they evaluate both of them in the RSNA dataset. Instead, the evaluation method in this paper is a way to normalize the scores between different classes by leveraging the negative mentions.

* **Evaluation**: VECL is designed specifically for the PNC evaluation. Therefore, the "zero-shot classification" is not truly "zero-shot" and the comparison is somehow unfair for other methods. A better way to evaluate the performance is to evaluate the quality of the backbone model by finetuning it in the downstream tasks, like the setting in [1]

[1] Huang, Shih-Cheng, et al. "Gloria: A multimodal global-local representation learning framework for label-efficient medical image recognition." Proceedings of the IEEE/CVF International Conference on Computer Vision. 2021.

**Questions:**

* **Evaluation**: What is the performance of the backbone besides zero-shot evaluation?

---

> ### Author Response · Authors · 2024-11-14
>
> About Motivation:
> All metrics in Gloria are POS evaluation method, which do not assess the distance between both negative and positive mentions. Only the PNC evaluation method can evaluate inter-class similarity, that is, the distance between images and positive as well as negative mentions.
>
> About Evaluation:
> VECL was not specifically designed for the PNC metric. In fact, CheXZero was the first to use the PNC metric, but it is a medical field paper and did not have much impact in the computer community. The mainstream work that followed still primarily used the POS metric. We are merely re-emphasizing the importance of the PNC metric, which has no relation to the significant advantages of the VECL method on the PNC metric.
>
> About Questions:
> The performance of the backbone is shown in Table 2, where we fine-tuned the pre-trained model with 1% of the data, and VECL also achieved state-of-the-art (SOTA) level.
>
> Did you really pay attention to the content of our article?

---

> ### Comment · Reviewer_jURR · 2024-11-14
> **Response to the rebuttal**
>
> Regarding Gloria (published in 2021), the zero-shot evaluation on RSNA (Table 3) involves both positive and negative mentions.
>
> Regarding the performance of the backbone. It's common to show the finetuned results (a finetuned version of Table 1) on the CheXpert and RSNA Pneumonia datasets as in the existing studies (for example, Gloria and MedKIP). Most datasets (3/4) used for zero-shot classification in Table 1 are missed in Table 2.

---

### Note · Authors · 2024-11-15

I have read and agree with the venue's withdrawal policy on behalf of myself and my co-authors.